# Improved Knee Function with Customized vs. Off-the-Shelf TKA Implants—Results of a Single-Surgeon, Single-Center, Single-Blinded Study

**DOI:** 10.3390/jpm13081257

**Published:** 2023-08-14

**Authors:** Peter Buschner, Ioannis Toskas, Jochen Huth, Johannes Beckmann

**Affiliations:** 1Clinic for Orthopaedics and Traumatology, Krankenhaus Barmherzige Brüder München, 80639 Munich, Germany; 2Sportklinik Stuttgart, 70372 Stuttgart, Germany

**Keywords:** customized, individually made, individual implants, knee arthroplasty, patient-reported outcome measures, activities of daily living, knee society score, Oxford Knee Score

## Abstract

Background: Recent studies have been able to show certain benefits of Customized, Individually Made (CIM) compared to Off-the-Shelf (OTS) total knee arthroplasties (TKAs), but evidence is still lacking regarding the benefits of these implant systems. This study aimed to find differences in scores and functional outcome by comparing CIM and OTS implants, using Patient-Reported Outcome Measures (PROMs) and functional tests for activities of daily living in a single-surgeon setup. Methods: A total of 48 patients (16 CIM vs. 32 OTS) were consecutively enrolled and blindly examined. Functional testing was performed using four timed functional tests (TUG, WALK, TUDS, and BBS) and the VAS for pain. The Aggregated Locomotor Function (ALF) score was then calculated based on the addition of the average times of the three functional tests. Results: The CIM group showed significantly faster times in all functional tests and significantly better ALF scores. There were remarkable differences in the assessment of maximum pain sensation between the two groups, with superiority in the CIM group. The PROMs analysis revealed a higher proportion of excellent and good ratings for the items objective and function (KSS) in the CIM group. Conclusion: The study showed that time-limited activities of daily living (ADLs) can be completed significantly faster with a CIM prosthesis and that a significantly higher percentage in this group reports freedom from pain during certain loads. Partial aspects of the PROM scores are also better in this group; however, this superiority could not be shown with regard to most PROM scores collected in this study.

## 1. Introduction

Today, total knee arthroplasty (TKA) is one of the standard procedures in orthopedic surgery. Like total hip arthroplasty (THA), it is characterized by impressive implant survivorship, but TKA has proven to be complex in terms of reconstructing natural kinematic patterns. This is reflected in lower patient satisfaction scores compared to THA, as well as compared to unicondylar knee arthroplasty (UKA) [1,2,3]. The latter does not change the kinematic patterns of the knee joint, therefore producing more satisfied patients [4,5].

Apart from various non-influenceable reasons that can affect the outcome and lead to patient dissatisfaction, such as individual expectations and health risk factors, the quality of care in recent years has focused on improving the implant design with regard to the kinematic aspects and creating a better fitting, as well as on new surgical techniques challenging the dogma of creating a neutral leg axis, which was propagated by Insall (1985) [6,7]. Above all of these innovations stands the heading “personalization”.

For example, various alignment strategies have been postulated to address this problem. The conceptual basis is to align the position of the prosthesis with the individual bony and soft tissue anatomy in order to ultimately—and this is common to all modern alignment strategies—avoid soft tissue releases as far as possible [8]. Furthermore, adjustments have also been made at the level of implant design. In addition to the proven cruciate retaining (CR) designs, where the posterior cruciate ligament is preserved, and posterior stabilized (PS) designs, where the posterior cruciate ligament is removed and its function is replaced by the design of the polyethylene (PE), medial-pivoting PEs or PEs that reconstruct an oblique joint line now also exist. Beyond that, a large number of asymmetric implants are available to improve the accuracy of fit. Moreover, the work of Bellemans and Hirschmann has sharpened our view regarding the different phenotypes on an anatomical and functional level, so it is logical that new technologies such as navigation/robotics, artificial intelligence (AI), and virtual and augmented reality (VR and AR) are entering the operating room, contributing to the individualization of care [9,10,11,12]. 

This ongoing personalization in the field of arthroplasty ultimately culminates in the production of customized instruments and implants by taking up the previously mentioned explanations and contributes to an individualized restoration with regard to implant design and positioning, whereby the optimal fit and position—in accordance with the surgeon’s favored concept—are already taken into account in the preoperative phase by means of sectional imaging, thus representing the highest form of personalization

Despite these developments, it has not yet been clearly demonstrated that customized implants provide an advantage in terms of patient satisfaction.

The actual literature concerning customized implants is controversial, despite the reasonable presumption of superiority of these implants. This is partly due to a lack of comparing studies [13,14,15,16,17,18,19,20]. Studies to date have not provided convincing evidence, particularly at the functional level, that customized implants are superior to conventional implants. This could be due to, among other things, the fact that the methods used to demonstrate better functionality are mostly based only on the collection of Patient-Reported Outcome Measures (PROMs) data, which are subject to ceiling effects in terms of their discriminatory power, especially in a younger and more demanding clientele [21,22,23,24]. 

To our knowledge, no study to date has combined functional tests for activities of daily living (ADLs) and PROMs data to demonstrate the functional superiority of these implant systems in a limited blinded setting.

The present study compared the clinical and functional quality of care of a cruciate retaining customized TKA with a series-produced total knee arthroplasty system with the goal of answering the following question:

Is there a difference in scores and the functional outcome when comparing patients with either Customized, Individually Made (CIM) or Off-the-Shelf (OTS) TKAs using PROMs data and functional tests for activities of daily living (ADLs) in a single-surgeon setup?

## 2. Materials and Methods

Patients were consecutively enrolled in the study and gave their informed consent; they were randomly selected solely by having underwent surgery 6 month prior. A comparison of clinical and functional outcome data of patients at a single follow-up visit at least 6 months postoperatively was performed. 

The surgery was performed by a single surgeon at a single center under the same surgical conditions as part of a multicenter study in the USA and Germany. The test administrator was blinded to the implants used. Objective knee outcome scores were used, which included functional tests and PROMs.

The inclusion criteria were (a) implant must have been in place for at least six months and free of complications, (b) subject received cruciate ligament preserving TKA, and (c) subject was >18 years of age. The exclusion criteria were (a) a simultaneous bilateral procedure; (b) a BMI > 40; (c) another physical disability requiring a walker or function-limiting disability of the hip, spine, contralateral knee, or other joint; (d) participation in another clinical trial that could affect results; and (e) inability, in the opinion of clinical staff, for safety or other reasons, to complete protocol.

Validated functional tests:

The Aggregated Locomotor Function (ALF) score [25]: obtained by summing the mean of the following three timed tests, which measure a person’s ability to perform various activities of daily living:Timed Up and Go Test (TUG) includes an ordinary chair, two meters of walkway, a time clock, and a leader who observes the subject and stops the time. The test leader instructs the proband to “walk” and begins to stop the time. The proband stands up from the chair, walks two meters, turns around, walks back to the chair, and sits down again. As soon as the proband’s buttocks touch the seat, the time clock stops. The proband is instructed to walk as fast as is safe and comfortable for him/her. Three timed trials are performed, the average of which is used for data analysis.The 8 m walk test (WALK) is a functional test used to measure locomotion ability. “Locomotion ability” refers to the activity of moving from one place to another, assessing walking ability, including walking distance, speed, and gait quality. In this test, the distance of four meters is marked with a pylon. The proband starts at one end of the walkway, walks at his/her normal walking speed to the pylon, around the pylon, and back to the starting point. Three timed trials are conducted, the average of which is used for data analysis.Timed Up and Down Stairs Test (TUDS): A time clock and stairs with a handrail are required for this test. The stairs consist of twelve steps with a height of 18 cm and a depth of 28 cm. The proband stands at the bottom of the stairs and, as instructed, climbs the stairs as quickly as is safe and comfortable for him/her and then turns around and walks back down the stairs. The test leader gives the command “Go”. At “Go”, the time clock begins to run. Once the proband is safely back down and has both feet on the landing, the time clock is stopped. The proband may hold on to the railing but should preferably perform this test without holding on to the railing. Two timed trials are performed, the average of which is used for data analysis.

The Berg Balance Scale (BBS) [26,27]: This is a performance measure of balance in the elderly. The Berg Balance Scale is used to identify elderly patients who are prone to falls. This scale includes 14 short tests that are scored by a test administrator. Each of the 14 short tests is based on a 5-point scale, ranging from 0 to 4. Zero indicates the lowest level of function, and four indicates the highest level of function. A fully functional patient has a maximum score of 56. The equipment needed consists of a ruler, two regular chairs (one with arms and one without), a footstool or step, a time clock, and a walkway of 4.57 m. The test administrator gives instructions to the proband while observing the proband’s ability to function.

Visual Analog Scale (VAS) for pain: In this test, the proband was asked to rate the pain in their knees during this test on a scale of 1–10 (1 is least; 10 is most).

Patient-Related Outcome Measures (PROMs): The 2011 Knee Society Score 2011 (KSS): The score consists of 4 separate subscales, (1) an objective knee score (seven items: 100 points), (2) a patient satisfaction score (five items: 40 points), (3) a patient expectation score (three items: 15 points), and (4) a functional activity score (19 items: 100 points). (For more detailed information, see https://www.kneesociety.org/assets/2011KSS%20Support%20Materials.pdf (accessed on 4 June 2023) [28]).

The Oxford Knee Score (OKS): The score is a questionnaire that consists of 12 items to assess function and pain after total knee arthroplasty. This results in a continuous score ranging from 0 (most severe symptoms/problems) to 48 (least severe). (For more detailed information, see https://innovation.ox.ac.uk/outcome-measures/oxford-knee-score-oks/ (accessed on 4 June 2023) [29]).

Implants: The ConforMis iTotal^®^ prosthesis (Billerica, MA, USA) CR variant was used as the patient-specific implant, which is based on a CT scan. The conventional implant used was the DePuy/Synthes P.F.C.^®^ Sigma (Warsaw, IN, USA) CR variant. The design of the prosthesis features a patent-specific fit in which it reflects the specific J-curves of the medial and lateral, as well as the patellofemoral groove. In addition, the implant features the incorporation of the patient-specific intercondylar offset and intercondylar width. CT data serve as the basis of the manufacturing process. The tibial component has patient-specific rotation and coverage.

The polyethylene is separated into a medial and a lateral part with different heights to mimic the natural joint line.

Surgical technique: In the OTS group, the implantation of the components followed the principle of mechanical alignment, where the components were implanted at 90 degrees to the mechanical femoral or tibial axis. In CIM, just the tibia is implanted at 90 degrees to the mechanical femoral or tibial axis. Individual joint obliquity, however, is incorporated in CIM in the femur, as well as the inlays of different heights, both medially and laterally. 

For the CIM implant, the extension and flexion gaps were prepared using patient-specific instruments and spacer blocks, and adjustments were made to the cuttings to balance the extension and flexion gaps if necessary. Femoral rotation was performed using a measured resection technique, with slightly optional adjustments to the rotation (up to 5°) according to ligamentous conditions. No releases other than those performed by osteophyte removal are necessary or carried out with CIM.

The OTS implant was positioned using conventional instrumentation, and the balancing of the flexion gap with regard to the rotation of the femoral component was based on the measured resection technique plus spacer blocks. Some balancing was carried out as needed and typical for the mechanical alignment technique.

Demographics: A total of 48 patients were included, randomly selected solely by having underwent surgery 6 months prior, and they were consecutively enrolled in the study. All patients were able to complete the clinical tests. The investigator who consecutively contacted and enrolled patients was blinded, thus accounting for the mismatch of group sizes. The details of the demographics are shown in Table 1.

The study was conducted in accordance with the Declaration of Helsinki and approved by the Institutional Ethics Committee of Landesärztekammer Baden-Württemberg, Germany, with the reference B-F-2015-098 (date of approval: 29 December 2015).

Statistical analysis was performed with IBM SPSS Statistics for Windows version 27 (IBM Corp., Armonk, NY, USA). Descriptive statistics are presented with mean and standard deviation (SD) for continuous variables, frequency counts, and percentages for categorical variables. Normal distribution was confirmed with the Shapiro–Wilk test, and the paired sample *t*-test was applied to determine the pre- and postoperative differences in continuous variables. Results are presented with a 95% confidence interval (CI), and a *p*-value of <0.05 was considered significant. 

## 3. Results

Functional tests: All functional tests (BBS, WALK, TUG, and TUDS) showed significantly (*p* < 0.01) faster times in the CIM group compared to the OTS group (Table 2). 

There were also remarkable differences in the assessment of maximum pain sensation when walking downstairs (Table 3).

PROMs: The KSS questionnaires revealed remarkable differences in the objective scores, as well as in the function scores (Table 4 and Table 5).

In the CIM group, 81.3% (13 probands) gave a rating of excellent and 12.5% (2 probands) gave a rating of poor, whereas in the OTS group, 59.4% (19 probands) gave a rating of excellent and 31.2% (10 probands) of poor.

The evaluation of the KSS questionnaires for the item function showed a rating of excellent (out of the four evaluation options: excellent, good, fair, and poor) by 62.5% (10 probands) of the patients with a CIM implant in contrast to only 37.5% (12 probands) of the patients with an OTS implant. In contrast, there was a rating of poor by 6.25% (one proband) of the patients in the CIM group and 25% (eight probands) of the patients in the OTS group.

However, the extended analysis of the KSS questionnaires revealed that no significant differences between the study groups were found in the scores for the KSS items satisfaction, expectation, and function, while a statistically significant (*p* = 0.03) difference was found for the KSS item objective (Table 6).

The statistical analysis of the five KOOS items, namely symptoms (*p* = 0.59), pain (*p* = 0.36), ADLs (activities of daily living) (*p* = 0.34), Sprt/Rec (sports/recreation) (*p* = 0.24), and QOL (quality of life affected by the affected knee) (*p* = 0.87), revealed no statistically significant differences between the CIM and OTS group (Table 7).

## 4. Discussion

The main finding of the present comparative study is the clinical and functional superiority of individual implants compared to conventional ones in a single-center, single-surgeon, and investigator-blinded study. 

The existing literature is conflicting. There are several studies that showed better clinical function after the implantation of an individual prosthesis [30,31,32,33]. However, a review from Müller et al. could not show significant benefits of CIM TKA compared to OTS TKA so far [34]. Several recent studies were not yet included, though [29,30,31]. Recent first studies with customized TKA of a second company with also partial constitutional alignment show promising results as well [35,36,37].

As commonly known, there is a discrepancy between preoperative expectation and postoperative satisfaction after the implantation of total knee arthroplasties [21,22]. Although the long-term survival rates of total knee arthroplasty are very satisfactory nowadays, there is a need for improvement in terms of achieving better satisfaction scores, especially considering that more and more young and more demanding patients are undergoing arthroplasty [23]. The younger patients are especially prone to having, and often have, worse results [21,23].

Dissatisfaction certainly has very different reasons and is therefore difficult to address, but one possible reason is the lack of anatomical accuracy of fit of some implants. In addition to the adaptation of conventional implants, such as improved size gradations, gender-specific implants, or the production of asymmetrical components, the use of individual implants offers the potential for improvement in this respect, as many studies have shown that there is a correlation between persistent pain and poorer functional outcome with implant mismatch, such as overhang, as well as malrotation [38,39,40,41,42,43]. 

A huge CT study by Meier et al., e.g., demonstrated that in patients with large AP and ML dimensions, the probability of mismatch of the femoral component with the bony anatomy was 25% of greater than 3 mm [44].

Furthermore, it seems to be obvious that an important goal should be the best possible restoration of anatomy to achieve better results [17]. 

Conventional implants are produced in incremental sizes, so fit is often a compromise between over- or under-dimensioning. This can be avoided with custom-made implants.

The manual determination of the implant size, as well as its positioning, is a source of error regarding the optimal fit and, thus, also the functional result. This was demonstrated by, among others, Thienpont et al., who showed that especially an increased external rotation of the femoral component leads to the measurement of a larger anteroposterior diameter and, thus, to a mismatch of the AP vs. ML dimension of the prosthesis [45].

To improve the fit intraoperatively, only the choice of using a smaller component is available. However, this means that compromises must be made. For example, reducing the size of the femoral component in an anteriorly referenced system leads to a loss of posterior offset and, thus, to a reduced flexion capacity or flexion instability. Conversely, anterior notching occurs with posteriorly referenced systems [45,46].

To achieve a symmetrical flexion gap, the femoral component usually is externally rotated using a mechanical alignment strategy. The use of a symmetrical implant, and particularly posterior referenced techniques, however, often leads to the loss of the medial posterior offset and, thus, to the creation of instability in midflexion [47]. The asymmetry of the posterior offsets of up to 6 mm in the natural knee joint was demonstrated by the work of Meier et al., who analyzed more than 24,000 CT scans. In the distal femoral, the offset/asymmetry was up to 8 mm. This wide variance of natural anatomy cannot be covered by even the asymmetric implants existing today [44].

This also accounts for the tibial side. In another CT study by Meier et al., of more than 15,000 CT scans, only 14% of the examined individuals had a symmetrical plateau with a difference medially to laterally of less than 2 mm, while 22% even showed more than 5 mm. When using symmetrical implants, this fact leads to a possible posterolateral overhang or distinct posteromedial under-sizing [48]. 

It is also worth mentioning in this context that the anatomical conditions influence and possibly affect the accuracy of fit through the choice of resection extent on both the tibial and femoral sides, as well as the rotation extent of the components to achieve balanced ligament tension and optimal patella tracking, so that conventional implants, although they are rich in variants, rarely provide optimal bony coverage [45].

The implantation of total knee joint endoprostheses is more prone to errors in achieving the planned position compared to the implantation of individual prostheses. Bugbee et al. showed that the use of conventional instrumentation leads to a greater deviation in the implant position compared to the planned, virtually implanted prosthesis. The use of custom instruments reduced the deviation from the planned position. 

Accordingly, the use of individualized instruments and implants helps to achieve the planned implant positioning and can, thus, have a favorable effect on function [49].

In this context, it seems quite obvious that the present study was able to show a functional superiority of the individual implants compared to conventional ones. This was shown to a statistically significant extent in the WALK and ALF tests. The reasons for this may be the optimized biomechanics and kinematics, as well as the improved fit of the customized prostheses. This improved function was also reflected in the objective score of the KSS. However, there were no significant differences in patient-reported satisfaction and functional activity. Similarly, we saw no significant difference when looking at the scores of the KOOS. 

Reimann et al., who also used the same CIM implant, found significantly better scores in the functional part of the KSS compared to an OTS implant [31]. This is in contrast to the present study, where the difference was only in the objective part of the KSS. Possibly, the difference is to be sought in the patient collective, as differences in demand also subjectively can judge the result differently and accordingly produce worse scores [50].

Regarding KOOS, other research groups also found no significant differences between the CIM and the OTS group. This can possibly be explained by the fact that complex activities are often not performed after total knee arthroplasty. In particular, this observation is supported by the fact that, in the subscale “sports”, both authors found the lowest score (<60 vs. 64) [31,51]. The same could be found in the present study, although the value here was slightly higher (67 points). Overall, however, few comparable works from the literature can be found that provide PROMs data. 

Generally speaking, a problem exists with ceiling effects concerning most PROM scores available and, consequently, their ability to detect differences between well-performing groups, because of clustering of participants’ scores at the upper limit of a scale; therefore, they have low discriminatory power among high-end scores [24].

The forgotten joint score or even short forms of existing PROM scores, such as KOOS-12, might be more effective, but the most objective or best combination of scores or even best or most objective score itself is part of actual debates [52].

As mentioned above, there is evidence that the implantation of unicondylar implants creates more satisfied patients [4,5]. Individual implants could fill this gap, although their superiority could only be shown to a limited extent in the analysis of PROMs data in the context of this study, possibly related to the ceiling effects described.

The used locomotor score (ALF), described and validated by McCarthy in 2004, is a score combining different activities and therefore presenting an easy and, through the combination of different activities, very objective score, with the single activities being presented as well [25]. It was further combined with the Bergs Balance Scale (BBS) in the present study to present an even more objective assessment of function, therefore further overcoming possible ceiling effects and revealing a clear improvement in functional results.

There are two companies actually manufacturing CIM, and both show very promising results [30,31,32,33,35,36,37,53,54,55]. However, a recent review could not prove its superiority, so far depicting the lack of good-quality studies [34]. Although several recent studies were not included in this review, a global recommendation of CIM cannot be stated until now.

Possibly, their potential in terms of improved fit, kinematics, and functional benefits might probably not be fully realized until the ideal fit is, e.g., combined with the ideal patient-specific alignment strategy and new scores with less ceiling effects are established. 

There are several existing limitations, including the low number of patients and difference in group size. However, it was a single-surgeon and single-center study, and the investigator was blinded who consecutively enrolled patients randomly who just had to be at least 6 months past implantation, accounting for this mismatch accordingly. The lack of preoperative PROMs data for comparison limits the assessment of the postoperative data, as the improvement potential of the respective implant design is not measurable. Unfortunately, and accordingly, the functional data in this study were not compared with a control group of the same age and normal knee joints to better evaluate their significance. Future studies should take this into account.

A comparison to other joint obliquity techniques cannot be drawn by using this study. Due to a variety of new alignment strategies that are increasingly coming into focus, a comparative study including those would also be useful in this context in the future.

Individual implants are used today, especially in cases with complicated anatomical conditions such as bone deformities or very small or very large anatomical conditions. In this respect, the collective treated with individual implants may have a special condition that is not directly comparable. The CIM collective was also statistically significantly younger than the OTS collective, so that this could also have influenced the result at the level of the PROMs. This, however, can also be seen as a strength of the study, since the endoprosthetic treatment of younger patients is known to limit the outcome [23]. As a causal factor for the younger age in the CIM group, there is basically the possibility of an unintentional selection. In addition, there is the possibility that patients of this group were better informed about and more open to alternative forms of treatment. Moreover, the likelihood of a later revision surgery, which is greater in the younger collective, might have led to the choice of a bone-saving surgical technique [56,57,58]. We further consider the locomotor score (ALF) being a combination of different activities and being further combined with the Bergs Balance Scale (BBS) to be a unique and clear strength of the study; however, since this method is uncommon, very few comparable data exist.

## 5. Conclusions

The present study found that CIM improves clinical and functional outcomes and is superior to OTS, not only in terms of accuracy of fit. Further prospective studies with larger case numbers and matched cohorts are necessary, though. The potential of CIM in terms of improved fit, kinematics, and functional benefits might probably be further pushed through combination with the ideal patient-specific alignment strategy and the establishment of new scores with less ceiling effects. 

## Figures and Tables

**Table 1 jpm-13-01257-t001:** Demographics.

	CIM		OTS		*p*-Value
	Average	Range	Average	Range	
# of Subjects	16		32		
Age (yrs.)	67	52 to 78	73	55 to 83	0.03
BMI	28	22 to 39	31	24 to 39	0.11
Post-op time (months)	13	6 to 25	18	7 to 45	0.12
Sex (male/female)	9/7		17/15		
Side (right/left)	8/8		17/15		

CIM: Customized, Individually Made. OTS: Off-the-Shelf. BMI: Body Mass Index, #: number of patients.

**Table 2 jpm-13-01257-t002:** Functional tests.

	BBS	WALK	TUG	TUDS	Normalized TUDS 10	Normalized TUDS 4	ALF Score 10	ALF Score 4
CIM	54	8.8	8.9	13.9	13.9	5.6	31.6	23.2
OTS	51	10.2	9.8	18.5	18.1	7.3	36.6	26.9
% Difference	6%	−14%	−10%	−25%	−23%	−23%	−14%	−14%
*p*-value	<0.01	<0.01	<0.01	<0.01	<0.01	<0.01	<0.01	<0.01

BBS: Berg Balance Scale. WALK: 8 m walk test. TUG: Timed Up and Go Test. TUDS: Timed Up and Down Stairs Test. ALF: Aggregated Locomotor Function. CIM: Customized, Individually Made. OTS: Off-the-Shelf. BMI: Body Mass Index.

**Table 3 jpm-13-01257-t003:** VAS pain.

Stairs Max. Pain	CIM	%	OTS	%
0	12	75.0%	18	58.1%
1	0	0.0%	3	9.7%
2	1	6.3%	4	12.9%
3–10	3	18.8%	6	19.4%

VAS: Visual Analog Scale. CIM: Customized, Individually Made. OTS: Off-the-Shelf.

**Table 4 jpm-13-01257-t004:** KSS objective scores.

	CIM	%	OTS	%
Excellent	13	81.3%	19	59.4%
Good	1	6.3%	3	9.4%
Fair	0	0.0%	0	0.0%
Poor	2	12.5%	10	31.2%

KSS: Knee Society Score. CIM: Customized, Individually Made. OTS: Off-the-Shelf.

**Table 5 jpm-13-01257-t005:** KSS function scores.

	CIM	%	OTS	%
Excellent	10	62.5%	12	37.5%
Good	2	12.5%	6	18.8%
Fair	3	18.8%	6	18.8%
Poor	1	6.3%	8	25.0%

KSS: Knee Society Score. CIM: Customized, Individually Made. OTS: Off-the-Shelf.

**Table 6 jpm-13-01257-t006:** KSS items.

	KSS Objective	KSS Satisfaction	KSS Expectation	KSS Function
CIM	91	32	11	80
OTS	77	33	10	73
% Difference	17%	−1%	14%	10%
*p*-value	0.03	0.92	0.11	0.15

KSS: Knee Society Score. CIM: Customized, Individually Made. OTS: Off-the-Shelf.

**Table 7 jpm-13-01257-t007:** KOOS items.

	KOOS Symptoms	KOOS Pain	KOOS ADL	KOOS Sprt/Rec	KOOS QoL
CIM	79	84	89	67	72
OTS	83	87	84	61	70
% Difference	−5%	−3%	5%	11%	2%
*p*-value	0.60	0.36	0.34	0.24	0.87

KOOS: Knee Society Score. CIM: Customized, Individually Made. OTS: Off-the-Shelf.

## Data Availability

No data available due to privacy and ethical restrictions.

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
