# Peer review of "Improved Knee Function with Customized vs. Off-the-Shelf TKA Implants—Results of a Single-Surgeon, Single-Center, Single-Blinded Study"

_jpm, 2023, doi:10.3390/jpm13081257_

Round 1

Reviewer 1 Report

In this study out of 48 patients received customized implants. The clinical outcome was studied and compared to the conventional group.

It is an interesting study, however it just lacks the numbers. Due to that missing power I cannot recommend publication.

In addition there are several other issues with the presentation of the manuscript such as length of abstract...

The language should be check... phrases such as "endoprosthetic replacement" are not commonly used 

Author Response

Overall, we thank for the valid comments. They clearly helped to improve the manuscript.

Answers are bold, changes highlighted

In this study out of 48 patients received customized implants. The clinical outcome was studied and compared to the conventional group.

It is an interesting study, however it just lacks the numbers. Due to that missing power I cannot recommend publication.

In addition there are several other issues with the presentation of the manuscript such as length of abstract...

The language should be check... phrases such as "endoprosthetic replacement" are not commonly used 

We appreciate your comment but consider the study definitively worth publishing being blinded and one of very few comparable studies concerning customized implants although having low numbers.

Length of abstract was edited, and language check was done. Please find the following line 11-27.

Reviewer 2 Report

This is a very good study which should be accepted for publication.

However there are some weaknesses in the presentation, which the authors should consider for an improvement of the paper.

First of all, the small number of patients. Although this cannot be dealt with without further work, the reader needs to be assured of the statistical analysis, so can this be made clear in the manuscript.

The authors deserve credit for using several functional tests which are clearly superior to the PROMS almost routinely used these days. PROMS very seldom show differences  between designs, techniques, robotic vs mechanical, and so on. Yet instinctively, there should be differences. The functional tests show definite relation to function which is an important criterion. Aspects such as satisfaction, used in PROMS, are vague and subjective.

In this study, did the authors have a set of functional data for normal subjects of the same age? That would be very valuable, to see how close are the total knee results to normal.

Another important consideration in this study is the differences between a custom TKA and an off-the-shelf (OTS). There are points raised in the Discussion, but nothing specific in the Materials and Methods in the specific features of a Custom knee which match it closely to the patients own knee in its normal state. This detail is needed. This point relates to the geometric features of the overall dimensions of the components, and also to the design of the bearing surfaces. The latter is specially important for the conformity and symmetry of the tibial bearing surfaces. Did The bearing resemble symmetric, or have medial-lateral asymmetry ?

Along the lines of the previous point, the surgical technique is paramount. There is a lot of evidence that Kinematic/Anatomic alignment produces better results and does not involve so much soft tissue releases. What was the situation with these series of knees, OTS and Custom? Was it exactly the same technique or a special technique for custom? How was the Balancing carried out in both series of knees? 

Author Response

Overall, we thank for the valid comments. They clearly helped to improve the manuscript.

Answers are bold, changes highlighted.

We appreciate the detailed and valid comments. They clearly helped to improve the manuscript.

This is a very good study which should be accepted for publication.

However there are some weaknesses in the presentation, which the authors should consider for an improvement of the paper.

First of all, the small number of patients. Although this cannot be dealt with without further work, the reader needs to be assured of the statistical analysis, so can this be made clear in the manuscript.

Thanks for the comment as this is a clear weakness of the study. Therefore, those small numbers are already mentioned in the discussion as a weakness of the study.

The authors deserve credit for using several functional tests which are clearly superior to the PROMS almost routinely used these days. PROMS very seldom show differences between designs, techniques, robotic vs mechanical, and so on. Yet instinctively, there should be differences. The functional tests show definite relation to function which is an important criterion. Aspects such as satisfaction, used in PROMS, are vague and subjective. 

In this study, did the authors have a set of functional data for normal subjects of the same age? That would be very valuable, to see how close are the total knee results to normal. 

We highly appreciate this comment. It is due to the limited blinding of the study (investigator did not know type of implant and patients solely been contacted and included according to time of surgery).

Unfortunately, and accordingly, the functional data in this study were not compared with a control group of the same age and normal knee joints to better evaluate their significance. Future studies should take this into account.

We added this to limitations. Please find the following line 573-576.

Another important consideration in this study is the differences between a custom TKA and an off-the-shelf (OTS). There are points raised in the Discussion, but nothing specific in the Materials and Methods in the specific features of a Custom knee which match it closely to the patients own knee in its normal state. This detail is needed. This point relates to the geometric features of the overall dimensions of the components, and also to the design of the bearing surfaces. The latter is specially important for the conformity and symmetry of the tibial bearing surfaces. Did The bearing resemble symmetric, or have medial-lateral asymmetry?

Again, we thank for this comment.

The design of the prosthesis features a patent-specific fit in which it reflects the specific J-curves of the medial and lateral as well as the patellofemoral groove. In addition, the implant features the incorporation of the patient-specific intercondylar offset and intercondylar width. CT data serve as the basis of the manufacturing process. The tibial component has patient-specific rotation and coverage.

The polyethylene is separated into a medial and a lateral part with different heights to mimic the natural joint line.

We added this to the paper. Please find the following line 347-353 and line 377-379.

Along the lines of the previous point, the surgical technique is paramount. There is a lot of evidence that Kinematic/Anatomic alignment produces better results and does not involve so much soft tissue releases. What was the situation with these series of knees, OTS and Custom? Was it exactly the same technique or a special technique for custom? How was the Balancing carried out in both series of knees? 

Important point. We thought that this was beyond the topic.

In the OTS group, the implantation of the components followed the principle of mechanical alignment, where the components were implanted at 90 degrees to the mechanical femoral or tibial axis. In CIM, just the tibia is implanted at 90 degrees to the mechanical femoral or tibial axis. Individual joint obliquity, however, is incorporated in CIM in the femur as well as the inlays of different hights medially and laterally.

For the CIM implant, the extension and flexion gaps were prepared using patient specific instruments and spacer blocks, and adjustments were made to the cuttings to balance the extension and flexion gaps if necessary. Femoral rotation was performed using a measured resection technique with slightly optional adjustments to the rotation (up to 5°) according to ligamentous conditions. No releases other than done by osteophyte removal are necessary or carried out with CIM.

The OTS implant was positioned using conventional instrumentation, and the balancing of the flexion gap with regard to the rotation of the femoral component was based on the measured resection technique plus spacer blocks. Some balancing was carried out as needed and typical for the mechanical alignement technique.

A comparison to other joint obliquity techniques cannot be drawn with this study and would be very interesting to be done in future.

Please find the following line 354-369 and line 377-379.

Reviewer 3 Report

Abstract

The authors should consider reducing the length of the abstract. Their abstract is longer than their Introduction section, which is a methodology flaw.

a. certain points are mentioned multiple times in your abstract. Just as an example - the statement "Analysis of KSS scores showed no difference between the two groups" appears twice.

b. Some sentences are not as clear as they could be, which could lead to confusion for readers - the phrase "solely by being six months after surgery" is not explicitly clear. This might come from a translation, and that might be the issue

c. There seems to be a contradiction between what authors sustained on the results and conclusion - the authors stated there were no differences in KSS scores between the groups, but in the conclusion, they claimed that the CIM group had better "objective" KSS scores

d. Ensure that all abbreviations are defined at first use. It appears that 'ALF' is used before its definition in the abstract. - this is applicable for the entire manuscript - please proofread your manuscript once more

e. Use of punctuation could be improved. For instance, you have a space before a comma in the sentence "23,2 and 31,6 s vs. 26.9 and 36,6 s, p <0.01". - this is applicable for the entire manuscript - please proofread your manuscript once more

f. "Partial aspects of PROMs scores (KSS "objective") are also better in this group..." could be rewritten for clarity... does this mean that the KSS scores were higher in the CIM group? Be clear and specific

Introduction

First of all, I recommend an expansion of your introduction section to provide a deeper background and context for your research.

a. There is a repetition of concepts - idea of individualized restoration, implant design, and positioning is mentioned multiple times - please revise

b. In the sentence "Actual literature is controverse...", 'controverse' should be 'controversial'

c. Some technical terms are not well-explained or defined - 'CR' and 'PS' designs are mentioned without proper context or definition - this is not an orthopedic journal exclusively. While we, as orthopedic surgeons, understand these abbreviations, many readers are outside this subject area.

d. The last sentence in the introduction ends with a period, but as it's a question, it should end with a question mark - please revise this sentence of objectives

e. In the introduction, you mentioned that satisfaction scores for TKA were lower than THA and UKA but did not elaborate on this point in the rest of the text - please provide more context.

Methods

Suggestions

a. Present the study design and inclusion/exclusion criteria at the beginning to offer a clear picture of the study population.

b. Discuss the surgical procedures used and the specifics of the implants. In addition, provide a rationale for the use of these specific implants.

c. A more comprehensive explanation of the statistical tests, their justification, and potential impact on the study results could help improve the understanding of the methodology

Results

I kindly recommend enhancing the Results section of your manuscript by including more descriptive sentences that elaborate on the findings. Seek a balanced proportion between textual explanations and the presentation of tables and figures, aiming for approximately 70% text and 30% tables/figures. This will provide a comprehensive and detailed account of your results, ensuring clarity and facilitating a better understanding for readers.

Discussions

well-structured and provides insightful analysis of the results. Additionally, the evaluation of the state-of-the-art is comprehensive and demonstrates a thorough understanding of the existing literature.

Conclusions

Effectively summarizes the key findings and their implications, providing a satisfactory closure to the study.

References

Ok

There are some grammar errors and some sentences seem to be translated. Also, please proofread everything regarding punctuation marks.

Author Response

Overall, we thank for the valid comments. They clearly helped to improve the manuscript.

Answers are bold, changes highlighted.

We very much thank for the detailed and valid comments. They clearly helped to improve the manuscript.

First of all, I recommend an expansion of your introduction section to provide a deeper background and context for your research.

Thank you for this important comment. We shortly added some details from the discussion into introduction (as according to Reviewer 2 we added more details into Materials and Methods section).

Please find the following line 31-161 and line 347-369.

  1. There is a repetition of concepts - idea of individualized restoration, implant design, and positioning is mentioned multiple times - please revise

We appreciate this comment and edited the manuscript accordingly limiting redundance and highlighting it in some parts. We checked and reviewed the introduction carefully to exclude repetitions.

Please also find comment above.

  1. In the sentence "Actual literature is controverse...", 'controverse' should be 'controversial'

Thanks, done.

  1. Some technical terms are not well-explained or defined - 'CR' and 'PS' designs are mentioned without proper context or definition - this is not an orthopedic journal exclusively. While we, as orthopedic surgeons, understand these abbreviations, many readers are outside this subject area.

We apologize for this which was edited accordingly.

  1. The last sentence in the introduction ends with a period, but as it's a question, it should end with a question mark - please revise this sentence of objectives

Thanks, done.

  1. In the introduction, you mentioned that satisfaction scores for TKA were lower than THA and UKA but did not elaborate on this point in the rest of the text - please provide more context.

We appreciate this comment. The mentioned statement is well known and several attempts (surgically, technically and implant wise) to near the results of TKA to THA or UKA have been tried. One is customized implants. So far, improvements could be made but no closing of the gap. Ceiling effects of conventional scores also have an influence. The results with the rarely used scores and tests in the study-in-hand show a clear improvement in functional results which is a clear advantage.

Please find the following line 538-560.

Methods

Suggestions

  1. Present the study design and inclusion/exclusion criteria at the beginning to offer a clear picture of the study population.

Thank you, done.

  1. Discuss the surgical procedures used and the specifics of the implants. In addition, provide a rationale for the use of these specific implants.

Important point. We thought that this was beyond the topic.

In the OTS group, the implantation of the components followed the principle of mechanical alignment, where the components were implanted at 90 degrees to the mechanical femoral or tibial axis. In CIM, just the tibia is implanted at 90 degrees to the mechanical femoral or tibial axis. Individual joint obliquity, however, is incorporated in CIM in the femur as well as the inlays of different hights medially and laterally.

For the CIM implant, the extension and flexion gaps were prepared using patient specific instruments and spacer blocks, and adjustments were made to the cuttings to balance the extension and flexion gaps if necessary. Femoral rotation was performed using a measured resection technique with slightly optional adjustments to the rotation (up to 5°) according to ligamentous conditions. No releases other than done by osteophyte removal are necessary or carried out with CIM.

The OTS implant was positioned using conventional instrumentation, and the balancing of the flexion gap with regard to the rotation of the femoral component was based on the measured resection technique plus spacer blocks. Some balancing was carried out as needed and typical for the mechanical alignement technique.

A comparison to other joint obliquity techniques cannot be drawn with this study and would be very interesting to be done in future.

The design of the prosthesis features a patent-specific fit in which it reflects the specific J-curves of the medial and lateral as well as the patellofemoral groove. In addition, the implant features the incorporation of the patient-specific intercondylar offset and intercondylar width. CT data serve as the basis of the manufacturing process. The tibial component has patient-specific rotation and coverage.

The polyethylene is separated into a medial and a lateral part with different heights to mimic the natural joint line.

The design of the prosthesis features a patent-specific fit in which it reflects the specific J-curves of the medial and lateral as well as the patellofemoral groove. In addition, the implant features the incorporation of the patient-specific intercondylar offset and intercondylar width. CT data serve as the basis of the manufacturing process. The tibial component has patient-specific rotation and coverage.

The polyethylene is separated into a medial and a lateral part with different heights to mimic the natural joint line.

We added the following line 347-369.

  1. A more comprehensive explanation of the statistical tests, their justification, and potential impact on the study results could help improve the understanding of the methodology

Important comment. However, as all tests are well established and as we added several parts to almost all sections, we consider the discussion of statistics beyond the topic of the manuscript.

Results

I kindly recommend enhancing the Results section of your manuscript by including more descriptive sentences that elaborate on the findings. Seek a balanced proportion between textual explanations and the presentation of tables and figures, aiming for approximately 70% text and 30% tables/figures. This will provide a comprehensive and detailed account of your results, ensuring clarity and facilitating a better understanding for readers.

We appreciate this comment and edited the section accordingly, so there is more text now to provide a better understanding of the graphs/tables.

Discussions

well-structured and provides insightful analysis of the results. Additionally, the evaluation of the state-of-the-art is comprehensive and demonstrates a thorough understanding of the existing literature.

Conclusions

Effectively summarizes the key findings and their implications, providing a satisfactory closure to the study.

References

Ok

Round 2

Reviewer 2 Report

The authors have seriously considered the Reviewer comments, and have given satisfactory responses.

Reviewer 3 Report

Changes have been made. Paper is eligible for publishing now